# Unforgettable Lessons from Forgettable Images: Intra-Class Memorability Matters in Computer Vision

## Abstract

We introduce intra-class memorability, where certain images within the same class are more memorable than others despite shared category characteristics. To investigate what features make one object instance more memorable than others, we design and conduct human behavior experiments, where participants are shown a series of images, and they must identify when the current image matches the image presented a few steps back in the sequence. To quantify memorability, we propose the Intra-Class Memorability score (ICMscore), a novel metric that incorporates the temporal intervals between repeated image presentations into its calculation. Furthermore, we curate the Intra-Class Memorability Dataset (ICMD), comprising over 5,000 images across ten object classes with their ICMscores derived from 2,000 participants' responses. Subsequently, we demonstrate the usefulness of ICMD by training AI models on this dataset for various downstream tasks: memorability prediction, image recognition, continual learning, and memorability-controlled image editing. Surprisingly, high-ICMscore images impair AI performance in image recognition and continual learning tasks, while low-ICMscore images improve outcomes in these tasks. Additionally, we fine-tune a state-of-the-art image diffusion model on ICMD image pairs with and without masked semantic objects. The diffusion model can successfully manipulate image elements to enhance or reduce memorability. Our contributions open new pathways in understanding intra-class memorability by scrutinizing fine-grained visual features behind the most and least memorable images and laying the groundwork for real-world applications in computer vision. We will release all code, data, and models publicly.

## 1 Introduction

As we experience the world, our memory selectively retains certain objects while overlooking others McDermott & Roediger (2018); Halamish & Stern (2022); Leshikar et al. (2012). For instance, among numerous flowers we encounter, a pale white daisy might fade from memory, while a striking violet osteospermum with intricate shapes remains vividly etched (**Fig. 1**). What factors make some flowers more memorable than others? To answer this question, we first introduce intra-class memorability which refers to the ability of instances within the same object class to be remembered. To delve into why certain items are more memorable than others, despite belonging to the same overall category, we conduct human behavior experiments using a continuous recognition task. In this task, participants are presented with a sequence of new and repeated images from the same class with delayed repetition and they are asked to identify those repeats. The proposed paradigm is novel compared to existing research, where continuous recognition tasks typically involve sequential presentations of images from different classes Bylinskii et al. (2015); Khosla et al. (2015); Dubey et al. (2015); Goetschalckx & Wagemans (2019); Kramer et al. (2023); Hovhannisyan et al. (2021).

To measure the memorability of individual instances, we introduce the Intra-Class Memorability score (ICMscore). Unlike prior evaluation metrics Goetschalckx & Wagemans (2019); Kramer et al. (2023); Isola et al. (2014); Akagunduz et al. (2019), our ICMscore incorporates average hit and miss rates for an item's repetitions, weighted by the time intervals between its initial and

Figure 1: **An object instance can be more memorable than another within the same class.** Intra-class memorability refers to the ability of instances within the same category to be remembered. First, we present example images for the five exemplar classes. Images are identified as low (Row 1) or high (Row 2) memorability based on results from human behavior experiments. We then use these high- and low-memorability images to train and test state-of-the-art AI models across four downstream computer vision tasks: memorability prediction, image recognition, continual learning, and memorability-controlled image editing.

repeated presentations. Missing a repeat after a short interval indicates low memorability (LM), while remembering a repeat after a long interval signifies high memorability (HM).

Due to the lack of intra-class memorability datasets, we contribute the Intra-Class Memorability dataset (ICMD). The dataset contains 10 classes spanning over 500 images per class with their corresponding ICMscore aggregated from the responses of 2000 participants in the human behavior experiments on continuous recognition tasks. To demonstrate the effectiveness of our ICMD, we train and test AI models on different subsets of ICMD images based on their ICMscores across four downstream computer vision tasks: memorability prediction, image recognition, continual learning, and memorability-controlled image editing. Our experimental results show that a memorability prediction model trained on ICMD captures representative features diagnostic for intra-class memorability. Moreover, surprisingly, while highly memorable images seem more salient and thus easier to remember, we found that these images actually impair AI performance in image recognition and continual learning tasks. Finally, we use images from ICMD with and without masked semantic objects to fine-tune the state-of-the-art image diffusion model Brooks et al. (2022) using LoRA Hu et al. (2021) to control image memorability. We highlight our key contributions:

**1.** We introduce the novel and important problem of intra-class memorability and provide insights into why certain instances are more memorable than others, despite belonging to the same category.

**2.** We challenge existing paradigms for collecting memorability data on naturalistic images by introducing a new human behavior experiment on intra-class memorability. In addition, we propose a novel memorability metric, the ICMscore, which accounts for the time intervals before a repeated image is identified in continuous image recognition.

**3.** We collect ICMscore from the responses of 2000 participants on 5000 images spanning across 10 object classes and contribute the first dataset on intra-class memorability, named as Intra-Class Memorability dataset (ICMD). To demonstrate the usefulness of our dataset, we introduce four computer vision tasks: memorability prediction, image recognition, continual learning, and memorability-controlled image editing.

**4.** Our results show that our memorability prediction model is capable of capturing key features for intra-class memorability. Surprisingly, while HM images are more salient and tend to be retained longer in memory, training AI models on these images can actually impair their performance in image recognition and continual learning tasks. Finally, trained on ICMD images with and without their semantic objects masked out, our fine-tuned state-of-the-art image diffusion model can enhance or reduce the memorability of a given image by manipulating their semantic regions.

## 2 RELATED WORK

Research in cognitive psychology shows that certain visual and contextual features, such as salient image characteristics, emotional valence, and semantic richness, significantly increase the likelihood of remembering specific objects Schmidt (1991). This suggests that memorability is a predictable and universal property rather than a purely subjective one Konkle et al. (2010); Bylinskii et al.

(2022); Isola et al. (2011). Building on this, Isola *et al.* Isola et al. (2011) define "memorability" as the probability that an image will be remembered after a single viewing. Subsequent studies have examined both extrinsic factors, such as category information, and intrinsic factors, such as visual features beyond category Khosla et al. (2015); Dubey et al. (2015); Hovhannisyan et al. (2021). Yet, these works have not disentangled the respective contributions of extrinsic and intrinsic factors.

To mitigate extrinsic influences, later studies restricted their focus to specific super-categories such as animals or indoor versus outdoor scenes Bylinskii et al. (2015); Goetschalckx & Wagemans (2019); Kramer et al. (2023); Lu et al. (2020); Borkin et al. (2013), or extended memorability analysis to videos across super-categories Cohendet et al. (2018a); Newman et al. (2020); Cohendet et al. (2018b). However, these approaches still mix images from different categories, limiting control over intrinsic features. To address this gap, we introduce intra-class memorability, which examines why some images are more memorable than others within the same category. We design a novel human behavioral paradigm specifically targeting this dimension.

Prior memorability datasets Isola et al. (2011); Bylinskii et al. (2015); Dubey et al. (2015); Goetschalckx & Wagemans (2019); Kramer et al. (2023), with LaMem Khosla et al. (2015) as one of the largest for deep learning-based memorability prediction, have relied on repeated-image recognition tasks, with scores typically derived from the proportion of correct detections and sometimes adjusted for false alarms Bainbridge & Rissman (2018); Goetschalckx & Wagemans (2019); Kramer et al. (2023). While temporal spacing between repeats is known to affect memorability Khosla et al. (2015), prior protocols often used random intervals, introducing uncontrolled variability. In contrast, our experiment standardizes time intervals and introduces the Intra-Class Memorability score (ICMscore), which jointly accounts for hit and miss rates across predefined intervals. We further contribute the Intra-Class Memorability Dataset (ICMD), comprising 5,000 images with ICMscores derived from responses of over 2,000 participants.

Studies have shown that highly memorable (HM) images can positively influence human cognition. For instance, GANalyze Goetschalckx et al. (2019) uses generative models to alter image attributes and modulate emotional or cognitive responses. Peng *et al.* Peng & Bainbridge (2024) demonstrate that memorable images increase engagement in retrieval systems and improve recall in storytelling and education. Other works Pataranutaporn et al. (2024); Siarohin et al. (2019) further reveal how AI-modified images and videos shape memory. These findings highlight the growing importance of memorability prediction Lahrache & El Ouazzani (2022); Squalli-Houssaini et al. (2018); Hagen & Espeseth (2023). However, existing predictors depend on both extrinsic and intrinsic features and cannot capture fine-grained differences within a class. In contrast, our model, trained on ICMD, achieves more accurate intra-class memorability prediction.

While memorability has been studied in human cognition, its impact on computer vision tasks remains underexplored. In particular, the link between image features critical for recognition and those that explain memorability is unclear, with implications for continual learning. Using ICMD, we provide initial evidence that memorability influences AI performance in recognition and continual learning. Finally, to enable direct manipulation, we fine-tune the pre-trained InstructPix2Pix model (Ip2p) Brooks et al. (2022) with LoRA Hu et al. (2021), enabling controlled manipulation of the memorability of specific object classes.

## 3 INTRA-CLASS MEMORABILITY DATASET(ICMD)

### 3.1 HUMAN BEHAVIOR EXPERIMENTS

In contrast to prior memorability datasets Bylinskii et al. (2015); Khosla et al. (2015); Dubey et al. (2015); Hovhannisyan et al. (2021); Goetschalckx & Wagemans (2019); Kramer et al. (2023), we introduce the Intra-Class Memorability Dataset (ICMD), which pairs naturalistic images with human memorability annotations across ten object categories. The categories—volcano, dog, phone, lemon, car, bird, palace, flower, jellyfish, and teapot—were selected to ensure clear semantic separation without overlap. ICMD comprises 5,000 target images curated from ImageNet Deng et al. (2009), evenly distributed across these classes.

**Fig. 2** illustrates the schematic of our human psychophysics experiments. In each trial, participants viewed a sequence of images, with both targets and foils drawn from the same category. Each image was displayed for 1,200 ms followed by a 1,600 ms blank screen, during which participants could

Figure 2: **Schematic illustration of the human behavior experiment on intra-class memorability.** A sequence of new and repeated images is presented. Each image appears for 1200 ms, followed by a blank screen for 1600 ms. Participants are asked to identify those repeats by pressing the space on the keyboard. Red arrows indicate the correct identification of the repeated images. Unique foil images are inserted into the sequence whenever there is no target image in that location.

press the spacebar to indicate recognition of a previously seen image. Reaction times were recorded across both periods. Before the main experiment, participants completed a guided demonstration to ensure task comprehension. During the experiment, immediate feedback was provided after each response (green tick for correct, red cross for incorrect). Target images reappeared after intervals of $t \in 8, 16, 24, 32$ images, sampled uniformly. These values were chosen based on a pilot study testing intervals from 2 to 64: intervals shorter than 8 led to near-ceiling performance (over 93% recall), while intervals longer than 32 were too difficult, with more than 34% of participants failing to detect any repeats. Each trial contained 100 unique targets, with foils filling the remaining sequence; each target and its repeat were shown only once per participant. **Sec. A** and **Fig. A1** further highlight differences between our setup and prior studies.

We recruited 3,263 participants on Amazon Mechanical Turk (MTurk) Turk (2012). To ensure data quality, participants were disqualified if they misidentified more than three consecutive foils as targets in a row or failed to respond within 60 seconds. After filtering, responses from 2,000 qualified participants remained, yielding 4,000 trials and 346,537 key presses, with an average of 85 key presses per trial. The mean false positive rate was 15.24%. All experiments were conducted with informed consent under protocols approved by our Institutional Review Board.

## 3.2 INTRA-CLASS MEMORABILITY SCORE (ICMSCORE)

To assess intra-class memorability, we introduce the Intra-Class Memorability Score (ICMscore), which measures the memorability of an image independently of its categorical information. Previous research typically computes memorability scores using hit rates and miss rates of memory recall Bainbridge & Rissman (2018); Goetschalckx & Wagemans (2019); Kramer et al. (2023). However, the interval $t$ between image repetitions significantly affects memorability Khosla et al. (2015). As shown in **Fig. A2** and **Sec. B**, memory accuracy declines with longer intervals due to memory decays and increasing cognitive loads over time Wixted (2004); Sikarwar & Zhang (2024). Therefore, we incorporate the lengths of these time intervals into the ICMscore calculation.

For each target $I$, we receive key press responses from $n$ participants. The correctness of $i$th participant's response is denoted as $x_i$, forming a response set $\{x_1, x_2, \ldots, x_n\}$ for image $I$. A response of $x_i = 1$ indicates a correct identification, while $x_i = -1$ indicates an incorrect identification. The interval associated with each response $x_i$ is represented as $t_i$.

Since human memory retention diminishes over longer intervals, a longer interval $t$ makes correct identification of seen images more challenging. Therefore, when a participant correctly identifies an image after a longer interval, we assign it a higher memorability score $s$ to reflect the increased difficulty associated with the larger $t$. For correct responses ($x_i = 1$), the memorability score $s_i$ is computed using a linearly increasing function: $s_i = x_i \frac{t_i}{T}$, where $t_i$ is the interval between two presentations of a target image, and $T = 32$ is the maximum interval. Conversely, as longer time intervals are more prone to incorrect identifications of repeated target images, errors during shorter intervals are more indicative of the image's memorability. For incorrect responses ($x_i = -1$), shorter intervals incur a larger penalty, while longer intervals result in a smaller penalty. The memorability score for incorrect responses is given by $s_i = x_i \frac{T-t_i}{T}$. The final ICMscore for each

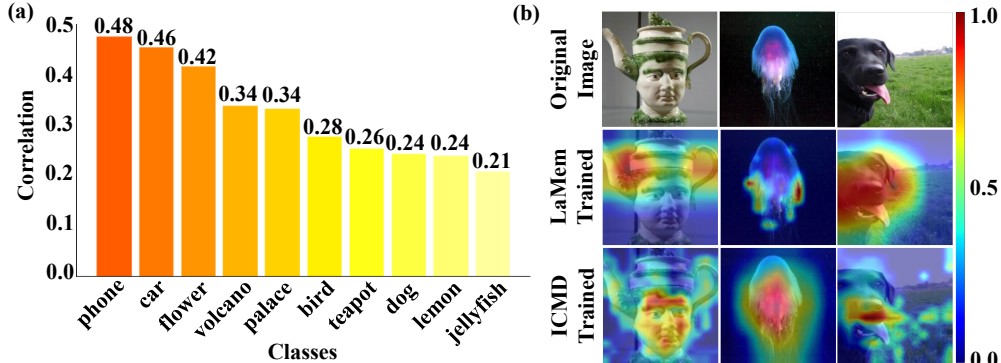

Figure 3: **Correlation analysis and Grad-CAM visualization.** (a) Pearson correlation between image feature distances to category centroids and their ICMscores. The y-axis indicates correlation values and the x-axis shows class labels, ranging from $r = 0.21$ for "flower" to $r = 0.21$ for "phone" ($p < 0.05$ for all classes). (b) Grad-CAM visualizations of three high-memorability (HM) images. Importance maps are derived from gradients of the memorability predictor with respect to feature maps, highlighting regions that contribute most to memorability scores. For HM images, memorability-enhancing features are highlighted according to the color bar on the right. The first row shows original images, the second row shows results from the LaMem-trained model Khosla et al. (2015), and the third row shows results from our ICMD-trained model.

target image $I$ is calculated as the average of its memorability scores $s_i$ over all $n$ participants: ICMscore(I)= $\frac{\sum_{i=1}^{n} s_i}{n}$.

To highlight the importance of the temporal penalty in our ICMscore formulation, we conducted an additional analysis by removing the temporal component $t_i$. Results show that without this penalty, ICMscore fails to capture intra-class memorability effectively, diminishing its utility for downstream computer vision tasks. See **Sec. C** and **Fig. A3** for details. Furthermore, to demonstrate that ICMscore is not merely a variant of existing memorability metrics defined in Bainbridge & Rissman (2018); Goetschalckx & Wagemans (2019); Kramer et al. (2023), we applied an existing memorability prediction model trained on LaMem dataset Khosla et al. (2015) to estimate memorability scores for images in our ICMD dataset. The two sets of scores show only a weak positive correlation, with a Pearson coefficient of 0.26. This suggests that ICMscore captures distinct memorability-related features overlooked by prior metrics and serves a complementary role in memorability evaluation.

### 3.3 DATASET ANALYSIS

Given the target images and their corresponding ICMscores, we now analyze how visual features of target images influence ICMscores. To investigate this relationship, we use a ResNet He et al. (2015) model pre-trained on ImageNet Deng et al. (2009) for object recognition tasks to extract feature representations for each image. We define the mean of all image features belonging to the same category as the centroid. Next, we compute the Pearson correlation Sedgwick (2012) between the L2 distances from each centroid to individual target image features and their ICMscores.

Our results reveal a positive correlation between an image's distance from its categorical centroid and its ICMscore across all object classes, as shown in **Fig. 3(a)**. The further away the features of an image are from its categorical centroid, the more memorable the image becomes. Here, we explore the connections between memorability and feature distance within categories. Insights gained from this intra-class feature analysis motivate us to design the experiments that investigate the role of memorability in the following computer vision tasks in **Sec. 4**.

### 4 EXPERIMENTS

#### 4.1 MEMORABILITY PREDICTION

Memorability prediction offers insights into human cognition and perception by revealing why some images are remembered more easily than others Perera et al. (2019); Harada & Sakai (2023); Needell & Bainbridge (2022). However, existing approaches conflate intrinsic and extrinsic factors. In contrast, our work predicts ICMscore by isolating intrinsic factors within the same category.

Specifically, we use a ResNet He et al. (2015), pre-trained on ImageNet Deng et al. (2009) for image recognition tasks, to extract image features, followed by two fully connected layers to predict memorability scores. We train our ICMscore predictor using the ICMD with an L2 regression loss, where the ICMscore of the input image serves as the ground truth. To validate the effectiveness of our predictor, we compare it against two baseline models. Baseline 1 employs the same network structure as our model but is trained from scratch without loading the pre-trained network parameters. For Baseline 2, inspired by previous work Hagen & Espeseth (2023); Constantin & Ionescu (2021), we use a Vision Transformer (ViT) pretrained on ImageNet Alexey (2020), append two randomly initialized fully connected layers, and fine-tune only these added layers while keeping the ViT backbone frozen. We perform 5-fold cross-validation to ensure the statistical robustness of the results. Following the practices in memorability prediction Isola et al. (2014); Hagen & Espeseth (2023); Khosla et al. (2015), we assess the models' performance by measuring the Spearman rank correlation Zar (2005) between the predicted and ground truth ICMscores. The first model achieves the highest correlation (0.64 ± 0.09), demonstrating that ICMscores can be effectively predicted using features extracted from pre-trained image recognition models, while the other two models obtained lower correlations: 0.60±0.2 and 0.57±0.05.

To verify that the 0.64 correlation from our predictor is not due to overfitting, we split participants into two independent groups and computed the Spearman correlation of their ICMscores on the same images. Averaged over 10 runs, the correlation was $0.69 \pm 0.05$, indicating high within-human consistency. While our model falls slightly below this human upper bound, it outperforms both baselines, suggesting it captures diagnostic visual features of memorability rather than overfitting. This result also aligns with the positive correlation between feature distance from category centroids and ICMscores (**Sec. 3.3**).

To further explore what image components contribute to high memorability, we employ Grad-CAM Selvaraju et al. (2017) to visualize importance maps for memorability, as shown in **Fig. 3(b)**. For highly memorable (HM) images, our predictor focuses on distinctive regions. For instance, in a teapot image with an ICMscore of 0.78, our predictor emphasizes the unique face-like texture of the teapot, while the model trained on LaMem Khosla et al. (2015) focuses on the teapot's spout and handle which are typical features reflecting the generic categorical appearance of teapots.

## 4.2 Image Recognition

We explore the role of memorability in image recognition by training models on image subsets with varying ICMscores. Images within each object class are sorted by their ICMscores and divided into two subsets: HM (High Memorability) with the top 250 images and LM (Low Memorability) with the bottom 250. Besides, we create a Mixed-Memorability (MM) subset of 250 images randomly mixing HM and LM images from the ICMD.

We employ ResNet He et al. (2015) as the backbone with random weights, and train it from scratch on each of the three curated subsets above over all ten classes using the Adam optimizer Kingma & Ba (2017). The initial learning rate is set to $1 \times 10^{-3}$ and reduced by 50% every five epochs over a total of 20 epochs following a linear decay schedule. The resulting models are labeled as HM-model, MM-model, and LM-model based on their training subsets.

We use the Cross-Entropy loss to train all three models for the 10-way classification task. Test images for object recognition are obtained from the original ImageNet Deng et al. (2009) test set regardless of their ICMscores. As these test images lack ground truth ICMscores, we could not evaluate the top-1 accuracy of the models on individual subsets of test images stratified by ICMscores. **Fig. 4(a)** presents the average top-1 accuracies of the three models on all the test images for all 10 object classes over 5 random seeds.

Our results in **Fig. 4(a)** indicate that the LM-model consistently outperforms the HM-model, achieving an average accuracy of 0.80 compared to 0.76 across all 10 object classes. Statistical analysis using one-way ANOVA reveal significant differences in top-1 accuracy between the two models ($p < 0.05$). Surprisingly, the LM-model also surpasses the MM-model. These findings indicate that image memorability impacts image recognition, with LM images improving accuracy and HM images impairing it. We also provide the confusion matrices for the classification performances of all three models in **Sec. D.1** and **Fig. A4**.

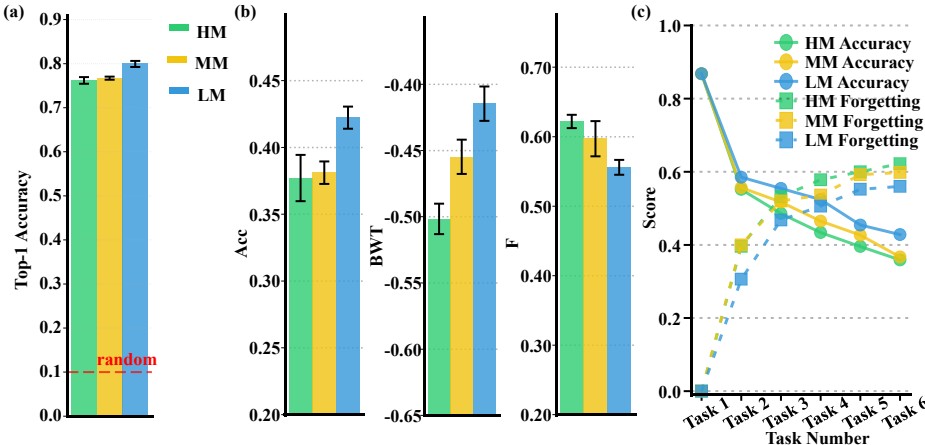

Figure 4: **Performance of AI models using HM, MM, and LM images for image recognition task (a) and continual learning with replay (b) and (c).** (a) Top-1 accuracy of image classification models trained on HM (green), MM (yellow), and LM (blue) images. The chance level for 10-way classification is 10% (red dashed line). Error bars denote standard errors over 5 runs. One-way ANOVA ($p < 0.05$) confirms significant accuracy differences among the three groups. (b) Continual learning performance of models replaying HM (green), MM (yellow), and LM (blue) images, evaluated by Accuracy (left), BWT (middle), and Forgetting (right). See **Sec. 4.3** for details. Error bars show standard errors over 5 runs. Two-tailed t-tests ($p < 0.05$) reveal significant differences between LM vs. HM and LM vs. MM across all metrics. (c) Accuracy at the last task (solid lines) and Forgetting (dotted lines) as a function of task number for continual learning models with HM (green), MM (yellow), and LM (blue) replay images. The y-axis shows either Accuracy or Forgetting, as indicated in the legend.

In real-world applications, training image recognition models in computer vision benefits from using images with varying memorability. However, obtaining memorability scores for all images through human behavioral studies is often time-consuming, costly, and therefore impractical. Therefore, a computational model for predicting an image's memorability becomes invaluable. We leverage the ICMscore predictor, introduced in **Sec. 4.1**, to estimate the ICMscore for all images and then use these predictions as criteria to conduct the same experiment for image recognition. See **Sec.D.2** for more details. From **Fig. A5 (a)**, consistent with the findings above, recognition models trained exclusively on LM images categorized by predicted ICMscores achieve higher top-1 accuracies than the MM-model. One possible reason is that easy images for object recognition often feature prototypical representations of object classes Singh et al. (2023), which accelerate training and foster robust, generalizable learning. As shown in **Sec 3.3**, easy images tend to be nearer to categorical centroids, and hence, less memorable but more beneficial for representation learning in AI models.

For comparison, we also conduct the same image recognition experiment based on the classical memorability definitions from prior works Bylinskii et al. (2015); Khosla et al. (2015); Dubey et al. (2015); Goetschalckx & Wagemans (2019); Kramer et al. (2023) to divide the dataset into two subsets and utilize the predictor from Khosla et al. (2015). The results, shown in **Fig. A5 (b)**, indicate that the predictor proposed by Khosla et al. (2015) does not enhance image recognition accuracy. Additionally, we observe no significant difference in top-1 accuracy between models trained on the HM and LM images as predicted by Khosla et al. (2015).

## 4.3 CONTINUAL LEARNING

Continual learning, particularly class-incremental learning, is essential for AI models to adapt to new classes over time without experiencing catastrophic forgetting of previously learned classes Parisi et al. (2019). One type of continual learning methods is based on memory replay Bagus & Gepperth (2021); Tee & Zhang (2023); Singh et al. (2023), which involves storing a subset of previously encountered examples in a memory buffer and replaying them during subsequent tasks. Given that some images are easier to remember while others are not, we conjecture that image memorability may play a key role in improving the continual learning performance of AI models. Here, we

investigate how memorability can be leveraged to select exemplar images for replay, potentially enhancing the model's ability to retain knowledge across tasks.

We structure our ICMD for continual learning by training object recognition models on the first 5 object classes in the initial task, followed by introducing 1 incremental class in each subsequent task. Since there are 10 object classes, this results in 6 tasks. To examine the role of image memorability, we further divide the images from each class into HM, MM, and LM subsets following the procedure in **Sec. 4.2**. During naive replay, we exclusively use images from one of these subsets, while keeping training images identical across all three continual learning models. The models replaying HM, MM, and LM images are denoted as HM-models, MM-models, and LM-models, respectively. We train each model for 10 epochs on the initial task and 5 epochs on each incremental task using Adam with a learning rate of $1 \times 10^{-3}$. The replay buffer size is fixed at 20 samples.

To benchmark the continual learning performance of these three models, we include the following three standard evaluation metrics, namely Accuracy at the last task (Acc), Backward knowledge Transfer (BWT), and Forgetfulness (F). We define $A_{t,i}$ as the accuracy of the model trained on all the tasks up to $t$ and tested on the test set of task $i$. Next, we introduce these metrics.

**Accuracy at the last task (Acc)** Shi et al. (2024) measures the accuracy of a model trained up to the final task $T$ and evaluated on all previous tasks; higher values indicate better continual learning. **Backward Transfer (BWT)** Lin et al. (2022); Díaz-Rodríguez et al. (2018); Shi et al. (2024) quantifies knowledge retention by evaluating performance changes on earlier tasks after learning new ones. **Forgetting (F)** Shi et al. (2024) denotes the accuracy drop on the first task when the model is trained to the last task. See **Sec. E.1** for details.

Results in **Fig. 4(b)** show that models using LM images for replay consistently outperform those using HM and MM images in Acc, BWT and F. We also plot accuracy and forgetting as functions of task numbers for all three models in **Fig. 4(c)**. The results indicate that the model replayed with LM images consistently exhibits reduced forgetting and achieves higher accuracy across all tasks. Together with our feature analysis in **Sec. 3.3**, our continual learning results align well with the previous research Rebuffi et al. (2017); Ke et al. (2024) that replaying samples where their features are closer to the categorical centroids yields better continual learning performance.

Similar to the image recognition experiments in **Sec. 4.2**, we categorize images into HM, MM, and LM using both our ICMscore predictor and the predictor from Khosla et al. (2015). From **Fig. A6**, there is no significant performance difference in replay methods using LM and HM images predicted by Khosla et al. (2015). In contrast, our ICMscore predictor remarkably reveals significant differences, with continual learning models replayed with LM images outperforming those with HM images or models using replay images predicted by Khosla et al. (2015) (see **Sec. E.2**).

We further conduct the above experiments on the prompt-based continual learning method Learning to Prompt (L2P) Wang et al. (2022) with an additional replay buffer. Consistent with previous results, we obtain the same conclusion that LM-based replay leads to superior performance. Full implementation details are provided in **Sec. E.2**.

### 4.4 Memorability-controlled Image Editing

Editing images with controlled memorability offers valuable applications across various research fields, such as designing advertisements for commercial products and improving students' knowledge retention ability in education Goetschalckx et al. (2019); Moran et al. (2019). We present a proof-of-concept demonstration of using a subset of teapot images in our ICMD to train an image diffusion model capable of modifying the memorability of an original teapot image.

To adjust a teapot image's memorability while preserving its semantic content, we train a high-memorability teapot generator (HM generator) using a subset of teapot images with high memorability (HM images) calculated from human behavior experiments (**Fig. 5a**). First, for each HM image, we use SAM Kirillov et al. (2023) to segment and mask the teapot area with white pixels. Next, these masked images are paired with their corresponding original images to train image diffusion models. Specifically, we fine-tuned the pre-trained InstructPix2Pix model (Ip2p) Brooks et al. (2022) with LoRA Hu et al. (2021). The generated images by Ip2p are compared against the ground truth HM images to improve its image generation quality.

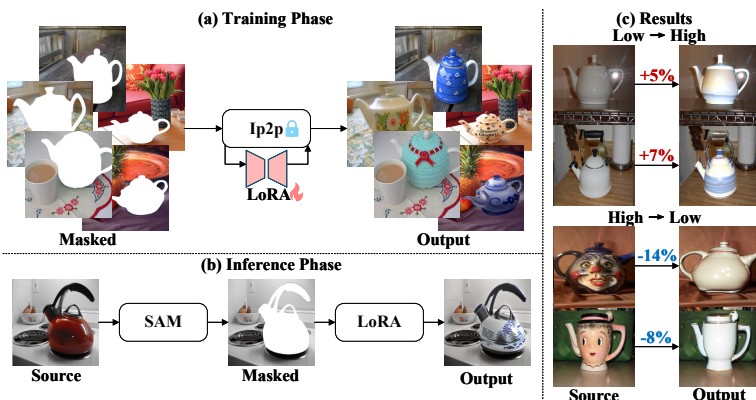

Figure 5: **Proof-of-conept demonstration of memorability-controlled image editing.** (a) Training: the image diffusion model Ip2p Brooks et al. (2022) is fine-tuned with LoRA Hu et al. (2021) using pairs of masked and ground-truth images. (b) Inference: SAM Kirillov et al. (2023) segments and masks the teapot region of a source image, and our fine-tuned model edits the masked image to generate teapots with varying memorability. (c) The high-memorability generator raises the ICMscore of LM teapots by adding vivid textures and increasing brightness, while the LM generator lowers the ICMscore by blurring distinctive features and simplifying textures. The percentage shown on each arrow represents the relative change in memorability score, computed as the difference between the edited and original image scores, normalized by the original score. All scores are computed using our memorability regressor in (**Sec. 4.1**).

During inference, the HM generator uses "highly memorable teapot" as the text prompt and the masked teapot image as input for image editing (**Fig. 5c**). The masked area guides the HM generator to focus on the specified region. The generator produces highly memorable teapot images. We qualitatively evaluate the HM generator's performance using LM teapot images from our ICMD. As illustrated in **Fig. 5c**, the generator effectively enhances memorability by adding visually striking textures to the original teapots. For quantitative evaluation, we use our memorability prediction model (detailed in Section 4.1) to predict ICMscores for both the original and generated images. The results indicate that the generated teapots consistently achieve higher ICMscores than the originals, demonstrating the effectiveness of our HM generator for memorability-controlled image editing.

We also employ a similar training strategy to create a Low-Memorability teapot generator (LM generator), with the results shown in **Fig. 5**. Similar analyses as applied to the HM generator are conducted. The findings reveal that memorability increases with the addition of distinctive visual elements, while it decreases when the teapot's unique textures are removed.

## 5 DISCUSSION

We introduce intra-class memorability, defined as the ability of individual instances within the same image class to be remembered by humans. To investigate why certain instances are more memorable than others, we contributed the ICMD dataset, comprising 5,000 images across 10 object classes. Memorability for these images is quantified using the ICMscore metric, derived from human behavior experiments in continuous object recognition tasks. To showcase the utility of ICMD, we train and evaluate AI models on subsets of the dataset, categorized by ICMscore, across four downstream computer vision tasks: memorability prediction, image recognition, continual learning, and memorability-controlled image editing. Our results reveal that intra-class memorability significantly impacts feature representation learning, with low-memorability images notably improving performance in image recognition and continual learning tasks. Building on our promising results, we outline several key directions for future research. While our study demonstrates memorability's role in computer vision using ten object classes, expanding ICMD to hundreds of ImageNet classes would provide broader insights. Furthermore, controlling image properties—such as standardizing backgrounds—could remove confounding factors and help isolate class-specific features critical for memorability. Although our findings highlight memorability's impact on computer vision tasks, its underlying mechanisms remain an open question. Lastly, while we focus on supervised training, future work could explore intra-class memorability in self-supervised learning frameworks.

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

## A    DIFFERENCES FROM PRIOR WORK IN HUMAN BEHAVIOR EXPERIMENTS

Previous studies typically present participants with sequences of images where each sequence contains images from multiple categories. However, image category can significantly influence memorability Bylinskii et al. (2015); Goetschalckx & Wagemans (2019); Kramer et al. (2023); Lu et al. (2020); Borkin et al. (2013). For example, dog images are generally more memorable than mountain images Kramer et al. (2023). To eliminate the influence of image categories on memorability, we design a new human behavior experiment focusing on intra-class memorability. **Fig. A1** compares our experimental protocol with previous approaches Bylinskii et al. (2015); Khosla et al. (2015); Dubey et al. (2015); Hovhannisyan et al. (2021); Goetschalckx & Wagemans (2019). Unlike previous approaches that mix categories within a single sequence, we present image sequences one category at a time. Each participant views sequences of object instances from only one category, with breaks inserted between categories to minimize cross-category interference in memorability.

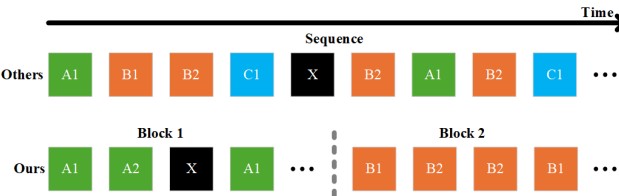

Figure A1: **The differences between our experimental protocol and previous studies.** Our experimental design differs from previous studies Bylinskii et al. (2015); Khosla et al. (2015); Dubey et al. (2015); Hovhannisyan et al. (2021); Goetschalckx & Wagemans (2019). In Row 1, "Others" represents the protocol used in previous studies, where participants are exposed to images from multiple categories within a single sequence. In Row 2, "Ours" represents our protocol, which requires participants to view only images from a single category per sequence, with the dotted line indicating the break between sequences. Different colored squares represent images from distinct categories, and the numbers indicate the position of each image within its category (e.g., B2 is the second image in category B). The black "X" represents a foil image that appears only once. In our protocol, the foil image "X" belongs to the same category as the target images in the same sequence.

## B    THE EFFECT OF TIME INTERVAL IN HUMAN BEHAVIOR EXPERIMENTS

In **Sec. 3.2**, we state that human memory retention tends to weaken over longer intervals. That is, a longer interval $t$ makes correct identification of the repeated images more challenging. See **Sec. 3.1** for the definition of interval $t$. We demonstrate this effect in **Fig. A2**. **Fig. A2** shows that identification accuracy of the repeated images and F1 score decrease with the longer interval $t$. Additionally, we conducted an experiment with $t$ exceeding 32, during which most participants struggled to complete the task. Therefore, we empirically set $t \in \{8, 16, 24, 32\}$ in real experiments.

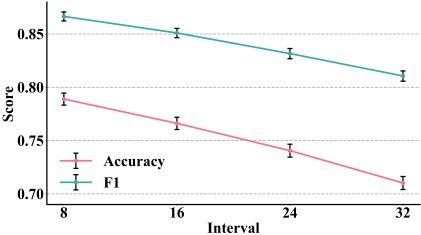

Figure A2: **Human behavior experiment scores.** This figure presents human performance at different intervals of $t$ (see **Sec. 3.2** for the definition of $t$). The x-axis represents $t$, and the y-axis represents the score, with the red line indicating the accuracy score and the green line indicating the F1 score Yacouby & Axman (2020). The values represent the average accuracy and F1 scores Yacouby & Axman (2020), computed across all trials with the same $t$. The error bars represent the standard error.

## C PERFORMANCE OF MODELS USING ICMSCORE WITHOUT TIME PENALTY

We modify ICMscore by removing the time penalty, defining $s_i' = 1$ (correct) and $s_i' = 0$ (incorrect), while keeping ICMscore(I)= $\frac{\sum_{i=1}^{n} s_i}{n}$. Following the dataset splitting protocol from **Sec. 4.2**, we train image recognition and continual learning models on LM, MM, and HM subsets.

**Fig. A3(a)** shows that removing the time penalty reduces the top-1 recognition accuracy by 2% compared to our best LM model introduced in **Sec. 4.2**. Similarly, the best continual learning performance in **Fig. A3(b)** degrades by 3% compared to our LM continual learning model introduced in **Sec. 4.3**. Overall, removing the time penalty leads to narrowing the performance gap among LM, MM and HM models in both object recognition and continual learning tasks. Notably, the key finding remains consistent: the LM subset helps train a better recognition model while mitigating forgetting.

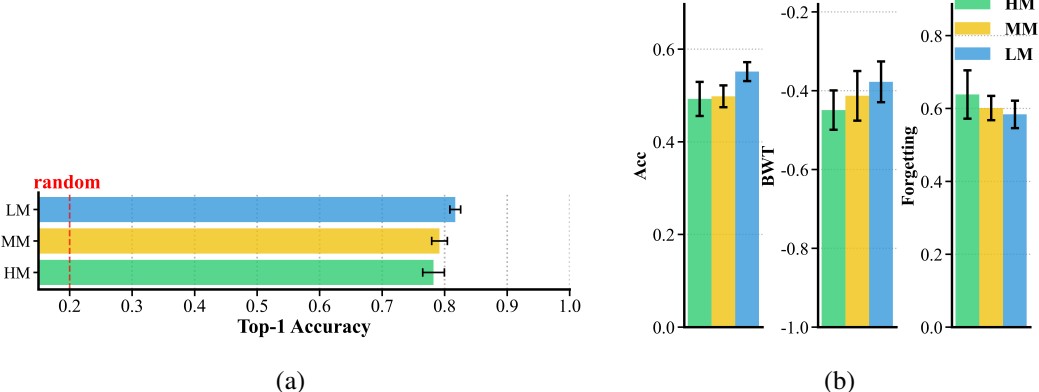

(a)                                       (b)

Figure A3: **Performance of Models Using ICMscore without time penalty.** (a) Top-1 accuracy comparison for image classification models trained on LM (blue), HM (green), and MM (yellow) images. The dashed line indicates the 20% chance level. (b) Continual learning performance showing Accuracy (left), BWT (middle), and Forgetting (right) for models using HM (green), MM (yellow), and LM (blue) replays. Error bars represent standard error over 5 independent runs. Evaluation metrics are detailed in **Sec. 4.3**.

## D IMAGE RECOGNITION

### D.1 CONFUSION MATRICES IN IMAGE RECOGNITION EXPERIMENTS

Following the image recognition experiment described in **Sec. 4.2**, we observe that models trained on LM images perform better in image recognition tasks than HM and MM images. To further demonstrate this difference, we compare the confusion matrices of models trained on three subsets. As shown in **Fig. A4**, the model trained on the LM images has a lower error rate across most classes, with the largest improvement in the "bird" class, where its accuracy increases by 16% and 15% in comparison to the HM and MM models.

### D.2 IMAGE RECOGNITION WITH OTHER MEMORABILITY PREDICTORS

We replace ICMscore in ICMD with the memorability score from prior work Khosla et al. (2015) and split the dataset into two subsets (highly and low memorable), following the procedure in **Sec. 4.2**. To examine the role of memorability in recognition, we sort images within each class by the memorability score and construct three subsets: HM (top 250), LM (bottom 250), and MM (250 randomly mixed from HM and LM). In comparison, we also apply our ICMscore predictor (**Sec. 4.1**) to form LM, MM, and HM subsets. These subsets are then used to train recognition models. **Fig. A5** reports performance across the LM-, MM-, and HM-models; see **Sec. 4.2** for discussion.

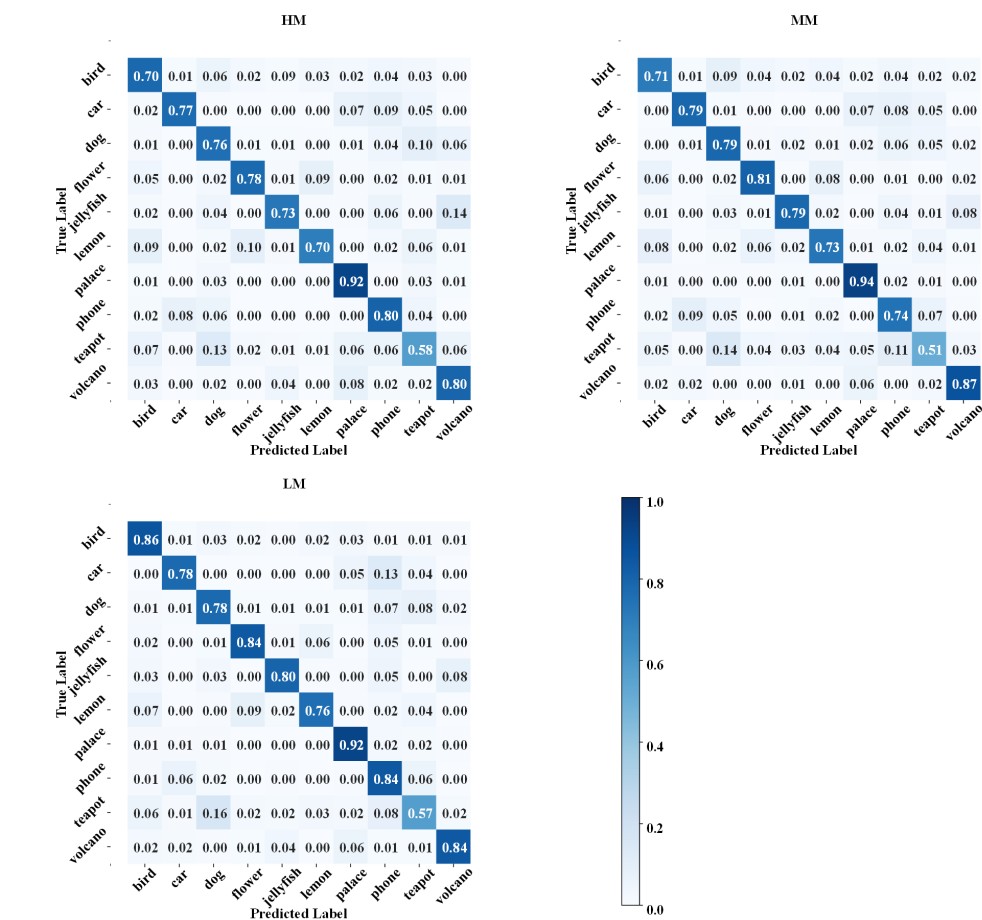

Figure A4: **Accuracy confusion matrices for models trained on HM, MM, and LM images classified based on human performances in behavior experiments.** Each confusion matrix shows the proportion of classifying images as predicted labels (columns) given the ground truth (rows). Values in all the entries of the matrices represent the average classification proportion over 5 runs. See the color bar on the right for the classification proportion.

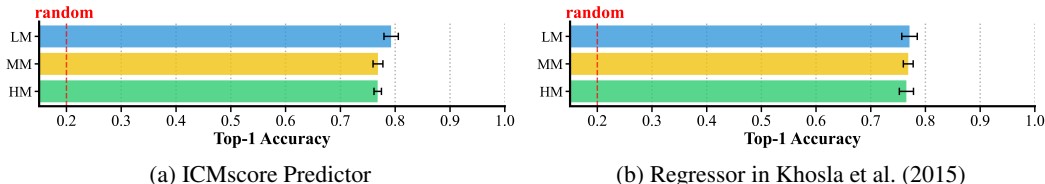

(a) ICMscore Predictor          (b) Regressor in Khosla et al. (2015)

Figure A5: **Performance comparison in Top-1 Accuracy between image classification models trained on HM, MM, and LM images using different memorability predictors.** (a) shows results using our ICMscore predictor from **Sec. 4.1**, while (b) presents results from the regressor proposed in Khosla et al. (2015). Within each subplot, from the top to the bottom, top-1 accuracy is reported for models trained on LM images (blue), HM images (green), and MM images (yellow). The chance is 20% for the 5-way classification (dash lines produced by the random model in red). The error bar is the standard error calculated over 5 independent runs with different random seeds.

# E    CONTINUAL LEARNING

## E.1    INTRODUCTION TO EVALUATION METRICS IN CONTINUAL LEARNING

To benchmark the continual learning performance of these three models, we include the following three standard evaluation metrics, namely Accuracy at the last task (Acc), Backward knowledge Transfer (BWT), and Forgetfulness (F). We define $A_{t,i}$ as the accuracy of the model trained on all the tasks up to $t$ and tested on the test set of task $i$. Next, we introduce these metrics in details.

**Accuracy at the last task (Acc)** Shi et al. (2024) is defined as the accuracy of the model trained up to the last task $T$ and tested on all the previous tasks. The higher the Acc, the better the model performs in continual learning tasks.

**Backward Transfer (BWT)** Lin et al. (2022); Díaz-Rodríguez et al. (2018); Shi et al. (2024) quantifies the model's knowledge retention ability by measuring performance drop on earlier tasks after learning new ones:

$$BWT = \frac{\sum_{i=2}^{T} \sum_{j=1}^{i-1} (A_{i,j} - A_{j,j})}{\frac{T(T-1)}{2}} \tag{1}$$

Typically, the BWT is a negative value due to catastrophic forgetting of the model. The lower BWT, the worse the model performs in continual learning tasks.

**Forgetting (F)** Shi et al. (2024) captures the accuracy drop on the test set of the first task for the model trained at the final task. It is defined as $F = A_{1,1} - A_{T,1}$. The lower the F, the better the model performs in continual learning.

## E.2    PERFORMANCE OF DIFFERENT CONTINUAL LEARNING MODELS

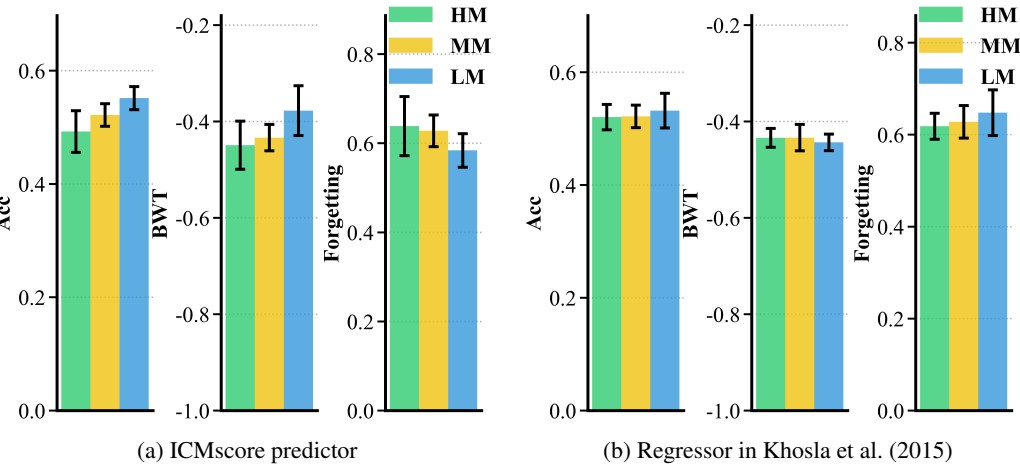

(a) ICMscore predictor                    (b) Regressor in Khosla et al. (2015)

Figure A6: **Continual learning performance of AI models using HM, MM, and LM images for replays.** (a) shows results using our ICMscore predictor from **Sec. 4.1**, while (b) presents results from the regressor proposed in Khosla et al. (2015). Within each subplot, from left to right, the performance of continual learning models using HM (green), MM (yellow), and LM (blue) images for replays is presented in Acc (left), BWT (middle), and Forgetting (right). See **Sec. 4.3** for the introduction to the evaluation metrics and the continual learning models. The error bars are the standard errors calculated over 5 independent runs with different random seeds.

Similar to the image recognition experiments in **Sec. 4.2**, we categorize images into HM, MM, and LM using both our ICMscore predictor and the predictor from Khosla et al. (2015). From **Fig. A6**, replay methods based on the predictor from Khosla et al. (2015) show no significant performance difference between LM and HM images. In contrast, our ICMscore predictor clearly reveals substantial differences: continual learning models replaying LM images consistently outperform

those replaying HM images, as well as models using replay images selected by the predictor from Khosla et al. (2015). See **Sec. 4.3** for a detailed discussion.

Beyond such naive replay strategies, we also evaluate more recent continual learning approaches. In particular, we adopt Learning to Prompt (L2P) Wang et al. (2022), which employs a small pool of key-value embeddings, or prompts, to encode task-specific knowledge while leveraging a frozen pretrained backbone. The input image feature representation is matched against the prompt keys via cosine similarity, and the top-$k$ prompts are prepended to the input embedding for downstream classification. To adapt this framework to our study, we extend L2P with a replay buffer that stores a random subset of previously seen examples. This design emphasizes the influence of memorability-aware replay data while disentangling its role during training. For the backbone, instead of the official ImageNet-pretrained ViT, we use a DINO-pretrained ViT Caron et al. (2021) to mitigate label leakage from ImageNet, ensuring a cleaner evaluation.

Following the same continual learning setup as in **Sec. 4.3**, we structure ICMD into 6 tasks by first training on 5 object classes and then sequentially introducing 1 incremental class at a time. Three variants of the augmented L2P model are trained, each using only HM, MM, or LM images for replay, while keeping the training images constant across variants. All models are trained for 10 epochs in the initial task and 5 epochs for incremental tasks, using a constant learning rate of 0.003. Other hyperparameter settings follow the original L2P configuration.

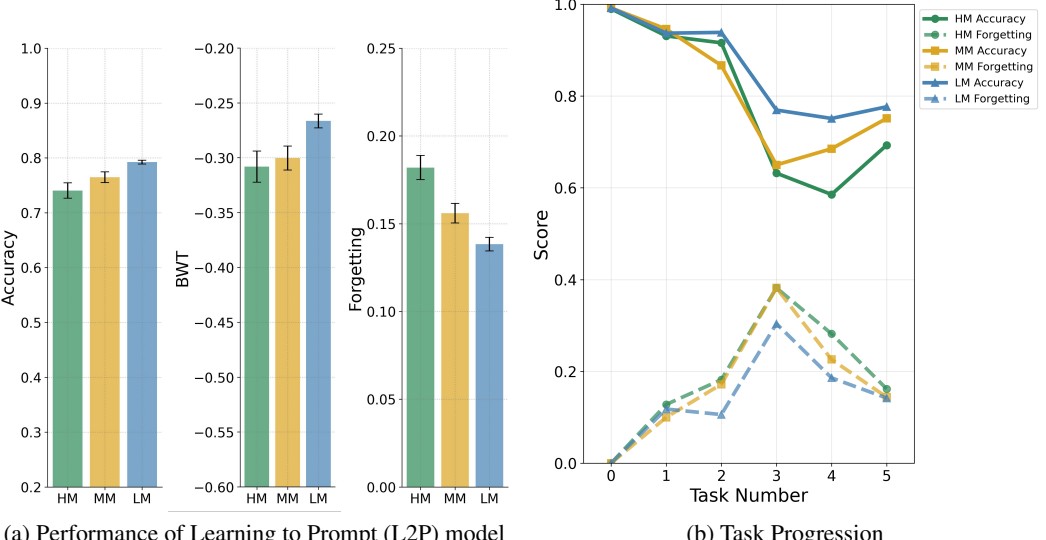

(a) Performance of Learning to Prompt (L2P) model    (b) Task Progression

Figure A7: **Continual learning performance of Learning to Prompt (L2P) Wang et al. (2022) augmented with memorability-based replay buffers**. (a) Performance comparison of L2P models that stores HM (green), MM (yellow), and LM (blue) images for the additional replay buffer, evaluated across Accuracy (left), BWT (middle), and Forgetting (right) metrics. Error bars represent standard errors calculated over 5 independent runs with different random seeds. (b) Task progression showing Accuracy at last task (solid lines) and Forgetting (dotted lines) as functions of task number for L2P models with HM (green), MM (yellow), and LM (blue) images stored in the replay buffer.

Results in **Fig. A7** show that L2P models using LM images for replay consistently outperform those using HM and MM images across all three metrics (Acc, BWT, and F), which aligns with the patterns observed in naive replay methods.

Finally, following the memorability predictor protocol in **Sec. 4.3**, we repeat the L2P experiments using both our ICMscore predictor and the regressor from Khosla et al. (2015). As shown in **Fig. A8**, the latter yields minimal differences between LM and HM replay, whereas our ICMscore predictor produces clear and consistent performance gaps. These results further highlight the advantages of memorability-based selection strategies.

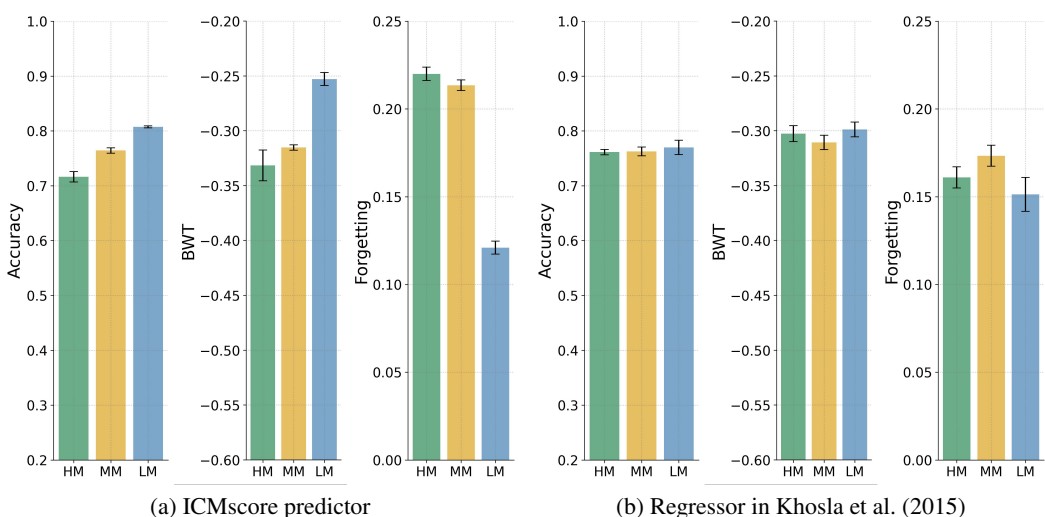

(a) ICMscore predictor  (b) Regressor in Khosla et al. (2015)

Figure A8: **Continual learning performance of Learning to Prompt (L2P) Wang et al. (2022) augmented with memorability-based replay buffers using different predictors.** (a) Results with our ICMscore predictor from **Sec. 4.1**. (b) Results with the regressor from Khosla et al. (2015). Metrics reported are Accuracy (left), BWT (middle), and Forgetting (right). Error bars indicate standard error over 5 independent runs with different random seeds.

