# OpenReview forum: "Unforgettable Lessons from Forgettable Images: Intra-Class Memorability Matters in Computer Vision"
_ICLR.cc/2026/Conference — ICLR 2026 Conference Withdrawn Submission_

### Official Review · Reviewer_UjRy · 2025-10-31

**Soundness:** 3
**Presentation:** 3
**Contribution:** 2
**Rating:** 2
**Confidence:** 4

**Summary:**

The authors investigate the properties and applications of how memorable an image to a human is. Contrasting with earlier works, they include a temporal dimension to memorability, i.e., if an image can be remembered even after a long time by a user, that should make it more memorable. The authors propose a new dataset of 5000 images and gather memorability scores with the aid of 2000 human subjects. They first show that memorability of an image (according to their score) is correlated with where that image is situated with respect to other images in the feature space of pretrained CNN models; if the image is close to the center, it has low memorability. The authors then use this score to partition images into high/low memorability and show that, perhaps counterintuitively, low-memorability images are more useful in training models for multiple computer vision tasks. They use image classification and continual learning as example. Finally, they use their highly memorable images to train diffusion models for image to image translation to make a boring image into a more exciting version (higher memorability).

**Strengths:**

- What makes an image memorable to a human person is a complex question. The steps taken by the authors over the previous work - (i) studying that property separately for each class, and (ii) adding a temporal dimension to it - are reasonable choices to make memorability more grounded.

- The scores are obtained over 2000 human subjects, which seems like a big enough sample size.

- The authors have empirically demonstrated that their proposed score is better suited to filter out the dataset for training models for downstream tasks, compared to memorability scores from existing methods. As an application, they can improve the accuracy of image classification models by training only on images with low-memorability score. Another application is shown for continual learning, where training on, again, low-memorability images leads to better learning trajectory.

- Finally, the authors show a toy-application of their method by training a diffusion model on memorable images, which can then translate a somewhat ordinary image (teapot) to a exotic version of it.

**Weaknesses:**

- The biggest limitation of this work is its ambiguous use case. There are two parts of the paper. The first concerns with what kinds of images are more memorable for humans, and whether there are any patterns in them, and why are they more memorable (this last question is not studied in this work). This part has more to do with cognitive science and (relatively) less directly applicable to the standard computer vision field. Nevertheless, I do not think we get any solid understanding about what makes images memorable. What the authors record is simply 'which images can humans remember'. But it does not answer why humans remember them. If those images are different, in what manner are they different? How exactly do they differ from 'important' images, or 'hard' or 'easy' images? I discuss this point in detail below.

- The second part of the paper is more application centric. However, in the image recognition experiments in Section 4.2, the total number of images used in each of the three models (LM, HM, MM) are 250*10 = 2500. This seems like a small dataset, especially for training a ResNet model from scratch (line 307). Consequently, it is difficult to know how this scales to bigger dataset or simply other kinds of datasets. A useful experiment for the authors will be to use their pretrained ICMscore predictor, and score images from a much larger dataset (e.g., ImageNet complete), and then train the model on only LM ImageNet vs HM ImageNet to see how those two models perform.

- Maybe most critical point to address is - how does the notion of memorability compare with other seemingly similar notions of "importance" (sample being important vs unimportant) [1], and "hardness" (sample being hard or easy) [2]. This will be important because the latter part of the paper is indeed centered around the application of memorable images. And these other related ideas are also designed for similar purpose; e.g., how to train a network more efficiently using just hard samples, or important samples.

- The final toy application of image to image translation feels a bit forced. First, it seems to break the flow of the previous section (image classification -> continual learning -> intructpix2pix image translation). Second, there are certain other baselines that you could try instead. To make an input image more 'memorable' you can give use the same text prompt that you use (“highly memorable teapot"; line 455) in the original, pretrained model. Maybe that can do a similar thing. Or change some words in the prompt (changing "memorable" to "beautiful" or "exciting").


References

[1] Not All Samples Are Created Equal: Deep Learning with Importance Sampling. Katharopoulos et al. arXiv 2019

[2] Contrastive Learning with Hard Negative Samples. Robinson et al. arXiv 2021.

**Questions:**

- Line 269: the authors should explain what intrinsic and extrinsic factors are.

- Line 273: “To validate the effectiveness of our predictor, we compare it against two baseline models” - it is not clear why the authors refer to one of the predictors as “our” predictor and the other two as “baselines”. All three models are just different variants based on architecture/initialization choices. It is not the case that the first model (“ours”) is some special design for the given task of ICMscore prediction. Therefore, the subsequent line 288 - “... our model … outperforms both baselines” is also a bit strange.

- Figure 3(b): I do not think there is something conclusive from these three examples. First, it is just three images. Second, even among these three, the case of dog (last row) is a bit confusing. It is not clear from the heatmap of the author's model what the important region is (I don’t think the nose of the dog is memorable). If the authors’ intention is to do a qualitative comparison of what different score predictors look at, they should do a more exhaustive analysis including many more examples.

Some pointers:

- There seem to be too many references to the appendix. Ideally, the paper should be mostly self sufficient with very limited references to the appendix.

- Line 353: words like “invaluable” in this context should be avoided, and should only be used in cases where the need of the proposed method/technique is, without doubt, critical.

Type: Line 229: it should be r=0.48 for “phone”

---

### Official Review · Reviewer_h9FW · 2025-11-01

**Soundness:** 2
**Presentation:** 3
**Contribution:** 1
**Rating:** 2
**Confidence:** 5

**Summary:**

This paper introduces intra-class memorability, the phenomenon where certain images within the same object category are more memorable to humans than others. To investigate this, the authors design a human behavioral experiment using a continuous recognition task confined to a single category, and propose a novel metric, the Intra-Class Memorability Score (ICMscore), which incorporates time intervals between image repetitions to quantify memorability.

They curate the Intra-Class Memorability Dataset (ICMD), containing over 5,000 images across 10 object classes, with ICMscores derived from responses of 2,000 participants. Using ICMD, they demonstrate that low-memorability (LM) images consistently improve performance in image recognition and continual learning tasks, while high-memorability (HM) images impair model accuracy and increase forgetting.

Furthermore, the authors fine-tune a diffusion model on masked–unmasked image pairs from ICMD to enable memorability-controlled image editing, successfully enhancing or reducing memorability by manipulating semantic regions.

**Strengths:**

1.	The paper conducts a thorough human evaluation and constructs an image memorability dataset, which may have positive implications for fields such as human-computer interaction.
2.	The writing is clear and well-structured, and the experimental evaluation is comprehensive.

**Weaknesses:**

1.	The paper’s central claim that highly memorable (HM) images impair model training is already a well-established consensus in the machine learning community.
2.	The experiments rely on an insufficient amount of data, which undermines their persuasiveness. Moreover, for some experimental results, it remains unclear what practical relevance or real-world applicability they possess.

**Questions:**

1. Regarding the core concept
a) HM images, as defined in this work, are essentially samples that deviate from the category centroid—effectively a form of outliers. From the perspective of data selection theory, machine learning benefits most from prototypical samples, i.e., the low-memorability (LM) images described in this paper. Therefore, I do not consider this observation to be significantly novel. Furthermore, I question the paper’s assertion that “Surprisingly, while HM images are more salient and tend to be retained longer in memory.” Once one understands the operational definition of HM images and examines the provided examples (e.g., Fig. 1 and 5), this conclusion appears entirely expected to any researcher familiar with computer vision. For instance, if there is noise in the dataset (such as the appearance of a cat image in the "car" class or a pure black image), then this sample obviously has high memorability, and its damage to model training is obvious.
b) A more compelling direction would be to investigate how far an image can deviate from its category prototype before it should be discarded. The current coarse partitioning into high/mid/low memorability is insufficient. Instead, the paper should analyze how different types of feature deviations (e.g., texture, shape, color—as seen in the diverse HM examples in Fig. 1 and Fig. 5) differentially affect memorability and model performance. Such an analysis would substantially enhance the paper’s scientific contribution.
2. Regarding the experiments
a) The experimental data scale is severely limited. For instance, in Section 4.2, images from only 10 classes are used to train a ResNet. This is inadequate to support a general claim about memorability’s impact on computer vision models, especially since the paper aims to establish a universal principle rather than address a narrow, task-specific scenario.
b) In Section 4.1, the reported Spearman correlation for ICMscore prediction is only 0.64. Given this modest performance, what justifies the claim that “ICMscores can be effectively predicted”? Have you tried to use the ICMscore predicted by the model to divide the samples instead of using human scoring?
c) The edited images in Section 4.4 (Fig. 5) appear markedly different from naturally occurring HM images (Fig. 1). The editing process seems capable only of modifying surface-level textures, not structural or shape-related attributes. From a signal processing perspective, this resembles the injection of high-frequency noise. It remains unclear how such edits relate to the deeper cognitive or representational underpinnings of memorability. Additionally, I am concerned that such edits might push images across semantic boundaries—e.g., could an edited basketball be misclassified as a football due to altered surface patterns?
d) What is the practical utility of the proposed memorability-controlled editing method? The paper should include follow-up experiments demonstrating that edited images, when used in downstream training (e.g., for recognition or continual learning), produce the expected performance trends (e.g., HM-edited images degrade accuracy, LM-edited ones improve it).

---

### Official Review · Reviewer_Ddwj · 2025-11-01

**Soundness:** 3
**Presentation:** 3
**Contribution:** 2
**Rating:** 6
**Confidence:** 4

**Summary:**

The paper introduces Intra-Class Memorability (ICM) quantifying memorability differences within the same category. The authors propose a new behavioral metric (ICMscore) that weights recognition accuracy by temporal distance, collect a large-scale dataset (ICMD, 10 categories × 500 images), and demonstrate that less memorable images facilitate learning and reduce catastrophic forgetting in computer vision models. The study also extends to memorability-guided diffusion editing to manipulate image memorability.

**Strengths:**

The move from category-level (as in MemCat) to intra-class memorability is a significant conceptual advance, isolating perceptual rather than semantic variability.
The one-category-per-session design eliminates inter-class bias, allowing finer-grained analysis of what makes specific instances memorable.
The findings link human memorability to machine learning performance (e.g., continual learning, image recognition), demonstrating cross-domain relevance.
Using generative models to control memorability represents a forward-looking and creative direction for both cognitive science and generative AI.

**Weaknesses:**

Since the recognition task is restricted to single-category sequences, participants likely adapt to the semantic scope over time.
This adaptation can lead to reduced discriminative load participants may shift from visual to semantic strategies, artificially inflating performance for later trials.

Moreover, as shown in [1] (despite the problem statement being different) temporal sensitivity to stimulus change decays when subjects engage with visually or semantically diverse stimuli. In the present work, short-term intra-class consistency could thus reflect an adaptation bias rather than inherent memorability differences.

This adaptation could also explain the lower human consistency scores reported here compared to prior inter-category datasets like LaMem or MemCat participants might be encoding category prototypes rather than image-specific features. I strongly encourage a discussion in this context.

Authors should include a time-segmented analysis of recognition accuracy to check for within-session adaptation effects, or randomize session category order to mitigate bias.

The diffusion-based editing pipeline (InstructPix2Pix + LoRA) produces synthetically altered images, which may deviate significantly from the training distribution of ICMD. Without human validation, it is unclear whether predicted changes in memorability scores reflect genuine perceptual memorability shifts or model artifacts. Similar to the approach of [1], a calibration between human and model memorability ratings for edited/OOD images is needed to verify that the editing model preserves semantic fidelity while altering memorability. Related work in generation should also be discussed.

**Questions:**

The authors’ temporal weighting (linear with repetition interval) lacks validation against psychophysical decay models (e.g., exponential, power-law). Can they explain this more clearly?

While the result that low-memorability (LM) images aid learning is intriguing, the causal link is underexplored. Are LM images simply more prototypical, leading to more stable internal representations? Or do they reduce overfitting by emphasizing category centroids? Can the authors give a more detailed discussion on this?

---

### Official Review · Reviewer_QbjG · 2025-11-02

**Soundness:** 3
**Presentation:** 3
**Contribution:** 3
**Rating:** 6
**Confidence:** 5

**Summary:**

This paper introduces the concept of intra-class memorability, addressing how certain images within the same object category are more or less memorable to humans, and how this property influences AI systems. The authors propose the Intra-Class Memorability score (ICMscore)—a novel metric that incorporates temporal spacing effects into memory measurement—and introduce the Intra-Class Memorability Dataset (ICMD), comprising 5,000 images from 10 object categories, with memorability scores collected from over 2,000 participants through controlled behavioral experiments.

The paper evaluates the ICMD dataset across four downstream computer vision tasks: (1) memorability prediction, (2) image recognition, (3) continual learning, and (4) memorability-controlled image editing. The main findings indicate that high-memorability images, while salient to humans, can actually degrade model performance in recognition and continual learning tasks, whereas low-memorability images tend to improve generalization. The authors also fine-tune an image diffusion model (InstructPix2Pix + LoRA) to manipulate image memorability.

Overall, the paper makes a novel and well-executed contribution that bridges cognitive psychology and computer vision, offering both a valuable dataset and new insights into how memorability interacts with AI models.

**Strengths:**

- The introduction of intra-class memorability as a measurable and distinct property from traditional inter-class memorability is conceptually innovative and well-motivated.
- The work successfully isolates intrinsic visual factors by controlling for class-level confounds.
- The ICMD dataset is significantly larger (10 classes, 5,000 images, 2,000 participants) than in prior iterations, providing sufficient diversity and statistical reliability.
- Clear justification is provided for interval selection in the memory task, addressing a common methodological gap in prior memorability studies.
- Incorporating temporal intervals into the memorability score formulation reflects a deeper understanding of human memory decay dynamics.
- The metric is mathematically transparent and validated against both behavioral consistency and alternative memorability predictors.
- The paper evaluates memorability effects across multiple vision tasks, demonstrating its influence beyond perceptual modeling.
- Statistical rigor is maintained (ANOVA, correlation, significance testing), and ablation analyses clarify the role of temporal weighting.
- The inclusion of Grad-CAM visualizations and feature-distance analyses offers interpretable explanations for why high-memorability images may impair model generalization.
- The manuscript is well-organized, clearly written, and visually improved compared to previous versions, with informative figures and consistent notation.

**Weaknesses:**

- Despite broad task coverage, the experiments rely mainly on ResNet and ViT architectures. Including comparisons with recent high-capacity models (e.g., CLIP, ConvNeXt, or vision-language transformers) could strengthen generality claims.
- The finding that high-memorability images impair model learning is compelling but still somewhat descriptive. A deeper causal or representational analysis (e.g., feature redundancy, overfitting to saliency) would improve theoretical grounding.
- The paper briefly discusses scalability but does not experimentally evaluate computational efficiency or domain transfer (e.g., to unseen categories or large-scale datasets).
- The framework’s practical extension to new object classes remains unclear—each new class currently requires human annotation.
- While comparisons with LaMem are included, more extensive benchmarking or discussion against datasets like MemCat, SUN-Mem, and recent machine memorability predictors (e.g., Han et al., 2023) would contextualize performance improvements.
- The memorability-controlled image editing experiment is promising but somewhat superficial. Quantitative results (e.g., fidelity, perceptual quality metrics) are limited, and the evaluation focuses mainly on qualitative examples.

**Questions:**

- How does the ICMscore generalize to new or unseen object categories without retraining or new human data collection? Have you explored zero-shot or transfer-learning variants of your memorability predictor?
- Can you provide additional analyses (e.g., representational similarity or saliency overlap) that explain why high-memorability images impair AI performance in recognition and continual learning tasks?
- Given the behavioral data collection and model training pipeline, what are the computational or practical limits for scaling ICMD to hundreds of categories? Could lightweight memorability prediction models be deployed on embedded systems?
- How consistent is the change in memorability (ICMscore) after editing across different object classes? Does the manipulation generalize beyond teapots or similar structured objects?
- The paper reports a weak correlation (r = 0.26) between ICMscore and LaMem predictions. Could this be further unpacked—e.g., which visual attributes contribute to divergence between the two memorability definitions?

---

### Note · Authors · 2025-11-22

I have read and agree with the venue's withdrawal policy on behalf of myself and my co-authors.